# Circulating HPV16 DNA in Blood Plasma as Prognosticator and Early Indicator of Cancer Recurrence in Radio-Chemotherapy for Anal Cancer

**DOI:** 10.3390/cancers15030867

**Published:** 2023-01-30

**Authors:** Agnieszka M. Mazurek, Ewa Małusecka, Iwona Jabłońska, Natalia Vydra, Tomasz W. Rutkowski, Monika Giglok, Rafał Suwiński

**Affiliations:** 1Center for Translational Research and Molecular Biology of Cancer, Maria Sklodowska-Curie National Research Institute of Oncology Gliwice Branch, Wybrzeze Armii Krajowej 15, 44-102 Gliwice, Poland; 2I Radiation and Clinical Oncology Department, Maria Sklodowska-Curie National Research Institute of Oncology Gliwice Branch, Wybrzeze Armii Krajowej 15, 44-102 Gliwice, Poland; 3Radiotherapy Department, Maria Sklodowska-Curie National Research Institute of Oncology Gliwice Branch, Wybrzeze Armii Krajowej 15, 44-102 Gliwice, Poland; 4Radiotherapy Clinic and Teaching Hospital, Maria Sklodowska-Curie National Research Institute of Oncology Gliwice Branch, Wybrzeze Armii Krajowej 15, 44-102 Gliwice, Poland

**Keywords:** anal cancer, radiochemotherapy, human papillomavirus, circulating tumor-related HPV16 DNA, plasma, viral load, SUVmax

## Abstract

**Simple Summary:**

Anal cancer, which is characterized by one of the highest human papillomavirus (HPV) positive case rates, is a rare disease accounting for only 0.5% of new cancer cases. Currently, there are no molecular markers for anal cancer. Circulating tumor–related HPV (ctHPV) in liquid biopsy as a substitute for conventional tumor biopsy offers great opportunities for survival prediction and molecular response assessment. Little is known about the ctHPV16 viral load in patients with anal cancer, its impact on survival, or its relationship to other clinical parameters. In this research we investigated the implication of ctHPV16 for monitoring treatment effects, disease relapse after treatment as well as impact on clinical outcome of anal cancer patients treated with definitive chemoradiation. We demonstrated the utility of detecting HPV status in addition to SUVmax and N, allowing for better stratification of the patient for therapy.

**Abstract:**

Background: Implementation of anal squamous cell carcinoma (ASCC) treatment modifications requires reliable patient risk stratification. The circulating tumor–related human papillomavirus type 16 (ctHPV16) may play a role in predicting survival or assessing treatment response. Methods: The study included 62 ASCC patients treated with chemoradiotherapy. A threshold of 2.5 was used to determine the maximum standardized uptake value (SUVmax). The ctHPV16 viral load (VL) was quantified by qPCR. Results: In the multivariate Cox analysis, lower SUVmax (*p* = 0.047) and ctHPV16–positive (*p* = 0.054) proved to be independent prognostic factors for favorable overall survival (OS). In the subgroup with the higher SUVmax, ctHPV16 and nodal (N) status were independent prognostic factors with *p* = 0.022 for ctHPV16 and *p* = 0.053 for N. The best survival rate (95%) presented ctHPV16–positive/N–negative patients. High ctHPV16 VL tended to be slightly specific for patients younger than 63 years (*p* = 0.152). The decrease in ctHPV16 VL to undetectable level after the end of treatment correlated with the overall clinical response. Conclusions: A prognostic stratification by SUVmax, ctHPV16 and N–positive status allows consideration of more aggressive treatment in high–risk patients (those with high SUVmax, ctHPV16–negative, and N–positive) or de–intensification of therapy in low–risk patients (those with low SUVmax, ctHPV16–positive and N–negative). However, prospective clinical trials on a large group are needed.

## 1. Background

The ongoing discussion on de–escalation or intensification of therapy for anal squamous cell carcinoma (ASCC) [1] includes novel scenarios for optimizing chemotherapy [2] or modification of radiation dose–volume [3,4]. Consideration of de–escalation in treatment of ASCC refers to the reduction of irradiated pelvic volume in order to minimize the risk of damage to normal tissue [5]. Appropriate risk stratification to identify high–risk patients, so that aggressive management can be implemented in this group, or low–risk patients who can be adequately treated with lower doses, may enhance the effectiveness and tolerance of combined treatment.

The importance of potential value of human papillomavirus (HPV) testing in the management and follow–up strategy of ASCC patients seems to be gaining momentum [6]. In the case of HPV–positive patients with ASCC, improved survival and a reduction in the rate of local recurrences were shown [7,8]. This was additionally confirmed by the results of the meta–analysis [9]. In the studies that used polymerase chain reaction (PCR) methodology to assess HPV status, overall survival (OS) was almost three times better in patients with those with HPV–positive tumors compared to HPV–negative tumors [9]. In addition to determining HPV status in tumor–derived tissue material, there is growing evidence that the use of liquid biopsy to determine HPV in the blood can provide valuable information on molecular response (MR) or molecular relapse [10,11].

Most of the studies on prognostic significance of HPV infection in anal cancer referred to results based on the analysis of tumor biopsy samples [12]. Studies on ctHPV16 (assessed for the blood) are still few and fragmentary [11,13,14,15], The researchers report that there was no difference in survival time between ctHPV–positive and ctHPV-negative patients (in pre–treatment samples). It was shown that only dividing by viral load allowed patients to be stratified in terms of survival prognosis [14,15]. Of note, the ability of early assessment of disease relapse based on the analysis of ctHPV16 is not robustly established due to rarity of disease and relatively low rates of relapses. There is a large gap in publications on the importance of ctHPV in anal cancer, while our prospective cohort study addresses several threads, among them survival, monitoring, and multivariable regression.

The hypothesis is that ctHPV16 derived from blood plasma may be useful for monitoring treatment effects, disease relapse and prognostic markers in curative radio–chemotherapy for anal cancer. Moreover, we hypothesize that it is possible to define, based on ctHPV16 analysis, clinical subgroups of low-risk patients who may, potentially, benefit from de-intensification of therapy and a high-risk group who may be candidates for more aggressive treatment.

## 2. Methods

The study included 62 patients with histologically–confirmed ASCC (squamous cell carcinoma of the anus) treated with chemoradiotherapy (CRT) at Maria Sklodowska–Curie National Research Institute of Oncology Gliwice Branch in 2012–2021. Patients have been treated according to RTOG 0529 protocol [16], consisting of two chemotherapy cycles with mitomycin C (MMC) and 5–fluorouracil (5–FU) and curative doses of fractionated simultaneous–integrated boost intensity–modulated radiation therapy (SIB–IMRT). Radiation therapy was planned based on positron emission tomography–computed tomography (PET–CT) scans. The tumor stage was defined according to the American Joint Committee on Cancer (AJCC 8th edition). Treatment response was evaluated according to the criteria for assessing response in solid tumors (RECIST) [17]. For delineation of the primary tumor SUVmax, threshold of 2.5 has been used in accordance with the institutional practice. Using the lower quartile, the median, and the upper quartile of SUVmax as cut–off values, 4 subgroups were defined: I ≤ 9.61; II 9.62–13.60; III 13.61–17.40; and IV ≥ 17.41. The project was approved by the Bioethics Committee at the Maria Sklodowska–Curie National Research Institute of Oncology Gliwice Branch, Poland. This study was approved by the local Bioethical Committee in accordance to the national regulations. Informed consent was obtained from all subjects involved in the study.

Peripheral blood (12 mL) was collected into K_3_EDTA tubes (Becton–Dickinson, Franklin Lakes, NJ, USA). Immediately after draw, blood samples were separated by double centrifugation (10 min at 4 °C, 300× *g* and 1000× *g*). Plasma was aliquoted and stored at –80 °C until DNA isolation. DNA was extracted with Genomic Mini AX Body Fluids kit (A&A Biotechnology, Gdynia, Poland) or automatically by Maxwell^®^ RSC (Promega, Madison, WI, USA) according to the manufacturer’s instructions.

Blood samples were collected from patients, during routine tests before treatment to estimate HPV status (*n* = 35), who were then monitored following chemoradiotherapy for up to 3 years. We also included 27 patients who had not been tested for HPV before treatment for post–treatment monitoring. The available samples and scheme of patient classification are summarized in Figure 1.

In this study, the term ctHPV16 (circulating tumor–related HPV16 DNA) refers to the detection of HPV16 DNA in the total circulating cell–free DNA (cfDNA) in plasma samples. To quantify ctHPV16 VL we used real–time PCR based on TaqMan technology. The oligonucleotides (probes and primers) were synthesized by Genomed S.A. (Genomed S.A, Warsaw, Poland). Amplification of human telomerase reverse transcriptase (TERT) was used as a marker of the total amount of genomic DNA present in samples. Each measurement consisted of standard curve (plasmid construct), negative control and a sample. All PCR reactions were performed using the Bio–Rad CFX96 qPCR instrument (Bio–Rad Laboratories, Hemel Hempstead, UK). ctHPV16 VL in plasma was expressed as a log_10_ of copy number of HPV16 DNA per 1 mL (log_10_ VL).

The statistical analyses were performed using the Statistica^®^ software program, version 13.0 (StatSoft, Dell Inc. (2016). Dell Statistica (data analysis software system), version 13. software.dell.com). The Mann–Whitney U test (two–tailed test) was used to compare the VL of ctHPV16–positive patients. Prevalence and frequency were presented as a number and percentage, and significance was expressed using Pearson chi–square. Cox proportional–hazards models were used to estimate hazard ratios (HRs) for OS and DFS. The backward elimination method was used for the selection of variables in the multivariate analysis of the Cox proportional hazards model. Overall survival (OS) rate was defined as the percentage of individuals in the group who survived after treatment. Disease–free survival (DFS) was defined as the length of time after treatment ends that the patient survives without any signs or symptoms of that cancer. The Kaplan–Meier curves were constructed for comparison of OS with the Peto–Peto test to estimate the *p*-value. The log–rank test was used to assess the equality of survival distributions across different strata. A *p*-value of less than 0.05 was considered to be statistically significant.

## 3. Results

### 3.1. Patients’ Characteristics

Women predominated in the analyzed group of ASCC patients; there were 49 women (79%) vs. 13 men (21%). The mean age of the patients was 63 years, median 63 years, and range: 19–83 years. Thirty (48%) patients were under 63 years of age and 32 (52%) were over 63 years of age. Five patients had been diagnosed with another malignancy between 6 and 21 years prior to treatment, 1 patient had breast cancer and colorectal cancer, 2 had cervical cancer, 1 had oropharyngeal cancer, 1 had breast cancer, 2 patients had genital warts. None of the patients received immunosuppressive therapy before or during treatment. The data about patients’ sexual life were not available. The following distribution was for the T classification: T1—5 (9%), T2—24 (41%), T3—23 (40%) and T4—6 (10%). The distribution for the N classification was: N–negative—36 (59%), N1a—17 (28%), N1c—8 (13%). The distribution for SUVmax classification was: SUVmax ≤ 9.61—14 (25%), SUVmax 9.62–13.60—14 (25%), SUVmax 13.61–17.40—15 (28%), SUVmax ≥ 17.41—12 (22%). There were 21 (38%) smokers versus 34 (62%) non–smokers. Assessment of the effect of ctHPV16 on treatment outcome and follow–up was limited to a subset of 35 patients who agreed to provide blood samples for this study, of which 8 (23%) were ctHPV16–negative and 27 (77%) ctHPV16–positive in the pretreatment sample.

### 3.2. Parameters Influencing Overall Survival in a Group of 62 Patients with ASCC

Five year actuarial overall survival (OS) for the entire ASCC cohort of 62 patients was 83% (95% CI, 73–94%). Out of 6 parameters, the N status (*p* = 0.006), SUVmax (*p* = 0.002) and sex (*p* = 0.041) appeared to significantly influence overall survival (Table 1, Figure 2).

The actuarial 5 year OS was 88% for women vs. 63% for men, 90% for those younger than 63 vs. 76% for those older than 63; 88% for patients with T1–T2 tumors vs. 76% for T3–T4 tumors; 93% for N–negative vs. 65% for N–positive disease; 89% for smokers and 83% for non–smokers; 100% for patients with SUVmax < 13.6 vs. 66% for patients with SUVmax > 13.6.

SUVmax, which was expressed in continuous values, appeared as a strong prognostic factor *p* = 0.014 (HR 1.13; 95% CI 1.02–1.25). Considering this finding, 4 subgroups were defined, I ≤ 9.61; II 9.61–13.6; III 13.6–17.4; IV ≥ 17.4, which appeared to be slightly less significant (*p* = 0.057).

Table 1 summarizes the prognostic impact of selected variables for OS of patients in the analyzed group, including the parameters of univariate and multivariate Cox regression model. In univariate analysis, N–positive status was a significant poor prognostic factor for OS (*p* = 0.016). In multivariate analysis, higher SUVmax (*p* = 0.018), N–positive (*p* = 0.023) and advanced T (*p* = 0.119) were poor prognostic factors for OS.

### 3.3. Parameters Influencing Overall Survival in a Subgroup of ASCC Patients with Known ctHPV16 Status in Pre–Treatment Sample

The analysis of pre–treatment plasma samples revealed 77% (27/35) cases to be ctHPV16–positive. A summary of patient characteristics for subset stratified by ctHPV16 status is provided in Table 2 column A.

Females tended to be more often positive for ctHPV16, compared to men (66% vs. 11%, *p* = 0.158). Furthermore, the multiple regression analysis of the relationship between ctHPV16 detection and clinical parameters (T1/T2 vs. T3/T4, N0 vs. N–positive, sex, cigarette consumption, age < 63, vs. age ≥ 63, SUVmax of 4 categories) indicated the significant relationship of ctHPV16 with sex (*p* = 0.041, beta coefficient = 0.382).

The actuarial 5 year overall survival (OS) for ctHPV16-positive patients with ASCC was 78%, and 75% for ctHPV16-negative. ctHPV16 did not appear to be a significant parameter affecting OS in the univariate analysis (HR = 2.2, *p* = 0.389, Table 3).

In multivariate Cox analysis, positive ctHPV16 test result was a good prognostic factor for DFS, although without significance (*p* = 0.096, Table 3). For OS analysis, high SUVmax (*p* = 0.047) appeared as a significant independent prognostic factors for poor OS, and, ctHPV16–negative tended to associated with worse prognosis (*p* = 0.054).

As tumors with a SUVmax less than 13.6 had a very good prognosis (Figure 2F, *p* = 0.002), additional analysis was performed in a subgroup of tumors with a SUVmax greater than 13.6. The multivariate analysis included sex, age, N and ctHPV16 status. In Cox multiple analysis, ctHPV16 and N status were independent prognostic factors (*p* = 0.022 for ctHPV16 and *p* = 0.053 for N). The best survival rate was assessed for ctHPV16–positive patients with N–negative disease who achieved 95% survival (Figure 3A). Patients with ctHPV16–positive and N–positive disease ranked second, with a survival rate of 63%. A worse survival outcome was for ctHPV16–negative patients, with 36% survival for those with N–negative, and 0% for those with N–positive disease (Figure 3A). Similar results we obtained for the analysis for advanced tumors (T3–T4), although ctHPV16 did not reach significance (*p* = 0.057 for ctHPV16 and *p* = 0.256 for N).

### 3.4. Implication of Pretreatment Viral Load (VL) of ctHPV16 in ASCC 

The results of the VL analysis for 27 ctHPV16-positive patients are presented in Table 2, columns F–H. The median of VL (N = 27) was 844 copies/mL with the range between 6 and 31,500. A Shapiro–Wilk test revealed that the distribution of the values was not normal (*p* < 0.0001). After conversion of the values to log_10_, the range between 0.8 and 4.5 were obtained with a normal distribution (*p* = 0.452). The following values of ctHPV16 have been obtained: median 2.92, mean 2.93, min 0.77, max 4.49, lower quartile 2.45, and upper quartile 3.60. There was no correlation between SUVmax and VL (r = –0.076, *p* = 0.711; r^2^ = 0.005). Some trend for correlation between VL and the age of patients were found (r = –0.312; *p* = 0.112; r^2^ = 0.097). Crude data (Table 2, column F) and log_10_ value (Table 2, column G) according to clinical parameters have been presented. The univariate analysis (Mann–Whitney U test) do not reveal any specific factor influencing the ctHPV16 VL (Table 2, column H). Multiple regression was performed for ctHPV16-positive patients to explain which factor affects VL levels (the log VL regression was assessed with the covariates of the following parameters: T1/T2 vs. T3/T4, N-negative vs. N-positive, sex, cigarette consumption, age < 63, vs. age ≥ 63, SUVmax. The multiple regression model showed that the many parameters affect a high VL: N-positive (*p* = 0.031), younger age of onset (*p* = 0.059), non-smoker status (*p* = 0.137) and female sex (*p* = 0.278).

### 3.5. Significance of ctHPV16 VL during Follow-Up of Patients with ASCC

Blood samples were taken at various stages of treatment during routine laboratory tests: at baseline (pre-CRT) (1–30 days before therapy); during therapy (CRT, after first chemotherapy cycle); posttreatment (end CRT) (1–30 days after CRT); three control (C) points during the follow-up: C 1–8 months, C 9–25 months and C 3 years after CRT. Despite the intermittent sample delivery that was caused by various patient- and hospital-related events, including the COVID-19 pandemic, in this paper we show the detectability of ctHPV16 as a potential biomarker for disease monitoring during and after treatment.

### 3.6. Monitoring of Patients with Tested ctHPV16 Prior to Treatment

The rate of decline in VL was individually dependent. Figure 3B shows the fluctuation in ctHPV16 levels during treatment and follow-up in 21 of 27 pretreatment ctHPV16-positive patients. For 19 cases, we observed a decrease in the ctHPV16 VL to the undetectable level at different time points. A decline in VL of that patients was connected with complete remission of the disease. By focusing on the CRT point, it was possible to assess a decrease to an undetectable level in 8 out of 10 patients. In 2 cases, ctHPV16 was still detectable. For # 25, a slight decrease in copy number from 2.8 to 2.2 (blue, Figure 3B) was associated with clinical progression and death. For # 17 (green, Figure 3B), a decrease in copy number from 3.6 to 2.8 was associated with partial regression, while a further decline in the 14th month was associated with complete disease remission. By focusing on end-CRT point, 6 patients had a decrease in ctHPV16 to 0 after the end of treatment, but in one patient (case #2), a slow decrease in the level of ctHPV16 was observed from 3.4 to 3.1 (red), which correlated with disease progression. In the following months, the patient did not have the ctHPV16 test, but died after 2.6 years.

### 3.7. Determination of ctHPV16 in Patients without Sample Collection before Treatment

Twenty-seven patients who had not been tested for ctHPV16 prior to treatment were also included in the follow-up. This made it possible to assess molecular recurrence. During monitoring, we detected ctHPV16 in the patient 24 months after the end of treatment. The detected molecular recurrence (presence of ctHPV16) correlated with clinical relapse. After adjuvant treatment of this patient, no virus was detected in plasma sample, which correlated with a clinical cure. In the remaining cases, no ctHPV16 was detected in the blood.

## 4. Discussion

In this study, we were able to identify, low and high-risk groups based on SUVmax, ctHPV16 and N status (Figure 3A). Such prognostic stratification may allow, in future clinical trials, consideration of more aggressive treatment in high-risk patients (e.g., high SUVmax, ctHPV16–negative, N–positive). Likewise, low-risk patients (low SUVmax, ctHPV16-positive, N-negative) may, potentially, benefit from de-intensification of therapy. However, such attempts will require new prospective clinical studies in sufficiently large groups of patients.

Chemoradiotherapy (CRT) has been considered the standard treatment for ASCC for over two decades, and concurrent chemoradiation with FU/MMC has a clinically meaningful impact on DFS and OS [18,19]. Our study included a homogeneous group of ASCC patients treated with 5-FU and mitomycin C–based CRT. Analysis of the entire cohort showed that high SUVmax and N-positive proved to be poor prognostic factors for OS, while an advanced tumor stage was an additional, but less significant, prognostic factor. Our results do not differ considerably from the other studies. Ajani et al. presented the highest impact of N-positive disease together with tumor size [20]. They suggested that the identification of patients with a tumor larger than 5 cm and N–positive status, who only have a 30% chance of being disease-free within 3 years, necessitates consideration of more intense therapeutic regiments. In the other study, they present the poorest survival rates and highest relapse rates for patients with T3–T4 and N–positive category disease [21]. Another challenge in prognostic stratification was the positron emission tomography (PET) assessment. It was shown that MTV (Metabolic Tumor Volume) of the primary tumor may be considered as a prognostic marker for OS in ASCC [22]. However, in multivariate analysis, apart from the MTV assessment (>7 vs. ≤7 cm^3^), the worse survival (and progression-free survival, PFS) was additionally influenced by the positive inguinal lymph nodes (in PET/CT) and the T3/T4 stage [22]. Kidd et al. have shown that SUVmax predicted an increased risk of lymph node metastases and poorer disease-free survival [23]. Although the results of clinical trials modified the TNM classification system, no other treatment was proposed for a selected group of patients with a higher risk category [23,24,25]. Interestingly, several issues remained unresolved. In the research of Gunderson et al., it was noted that a subset of N–positive/T2 patients had a similar or better prognosis than the N–negative/T4 (T4N0). Authors conclude that treatment of the group of patients with the highest risk of relapse is currently a challenge [21]. Of additional interest is the increase in the number of N–positive patients (7% per decade), which correlated with a better prognosis for both N–positive and N–negative patients, while the T–stage distribution remains constant [26]. While this can be attributed to novel imaging and treatment modalities, the increase in HPV-related cancer could potentially be another factor. In our study, we underscore that ctHPV16 appears to have a greater impact on prognosis than N status.

Recently, there has been a strong emphasis on the use of HPV testing due to better treatment outcomes in HPV-positive patients’ compared to HPV-negative oropharyngeal cancer (OPC) patients [27]. The prognostic importance of HPV was also shown in the group of patients with ASCC. In Mai et al.’s study, HPV-positive patients (determined as DNA HPV–positive/p16–positive) had the best prognosis, whereas HPV–negative patients (DNA HPV–negative/p16–positive or DNA HPV–negative/p16–negative) showed the worst outcome [28]. A meta-analysis showed that HPV DNA–positive or p16–positive ASCC had a better prognosis than their negative counterparts [9,29]. Meanwhile, a new approach to molecular diagnostics based on liquid biopsy is gaining great support, including ctHPV detection in the blood of ASCC patients [10,11].

The search for factors significantly influencing ctHPV (or VL) in ASCC patients so far has not brought unequivocal results. Significantly higher VL levels in patients with lymph node metastases compared to N-negative patients were reported only in Cabel et al. [13], while in our study and Bernard-Tessier et al. [14] pre-treatment ctHPV levels did not correlate with any characteristics of the patients.

On the other hand, several studies point to the importance of ctHPV in the survival of patients with ASCC (overall or disease-free). Residual ctHPV detected at chemotherapy completion was associated with shorter post-chemotherapy PFS [13,14] and a reduction of 1-year overall survival rate [14]. However, pre-treatment ctHPV had no significant prognostic effect [13]. In study of Bernard-Tessier et al., for prediction analysis, ROC (*receiver operating characteristic)* curves were performed for cut–off assessment and the level below the cutoff (2940 copies/mL) was associated with a longer PFS [14]. This baseline ctHPV was associated with a sensitivity of 67%, a specificity of 70%, and a positive predictive value of 80% and a negative predictive value of 54% to predict progression after chemotherapy. At this point, it should be emphasized that the qPCR methodology is a very sensitive method and determining the CV (coefficient of variation) to determine the precision (repeatability and reproducibility) and acceptance conditions is a crucial problem. An interesting approach was shown by Lefèvre et al.; three elimination patterns of ctHPV were observed during CRT that correlated with the outcome: fast responders with no local or distant failures; slow responders with high risk of local failures, no distant failures; persistent molecular responders with high risk of distant failures, but no local failures. In this way, patients with a significantly different risk of failure can be identified [15].

Our work sheds new light on the importance of nodal advancement in ASCC. It is possible that the high nodal stage in ctHPV-positive tumors may be the result of HPV infection. This phenomenon is highlighted by research on the OPC. Enlarged lymph nodes in the neck are very common in patients with HPV-positive oropharyngeal cancers [30]. Another interesting clinical observation is the high nodal grade in patients with small tumors (T1–T2) [31,32]. Through our study, we point out that HPV should be included in the stratification of patients with ASCC (as it happened in OPC in 2018), because HPV, by modifying the N feature, strongly influences the survival prognosis.

Our study has several limitations. The study was retrospective. The sample was relatively small and a small number of events occurred. In addition, the study was conducted in a single institution. Since the literature on the subject is quite limited, this paper aims to provide new data on this topic. We attempted to define a clinical subgroup of low-risk patients who may potentially benefit from de-intensification of therapy, and a high-risk group who may be eligible for more aggressive treatment. Through our multiple regression studies, we also showed that ctHPV16 is not the only significant predictor, as it should be noted that node staging is also of particular importance for larger tumors.

## 5. Conclusions

In summary, we present here the possibilities of performing a simple test for detecting the HPV genotype 16 in the blood (ctHPV16) for both primary diagnosis and for monitoring the responses to treatment. Moreover, we demonstrate that the prognostic significance of such a test can be enhanced by consideration of SUVmax and the N status. This creates perspectives for improved stratification of patients with regards to the intensity of treatment and the perspectives for the enriched ability for early detection of treatment failure and, thus, for the early salvage interventions.

## Figures and Tables

**Figure 1 cancers-15-00867-f001:**
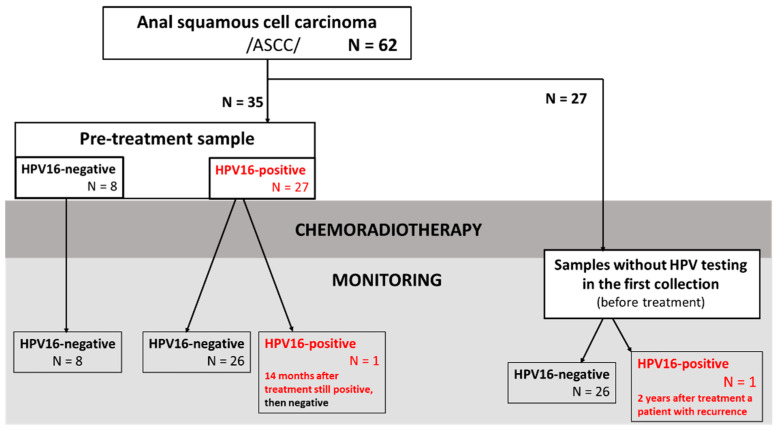
Scheme of availability of blood samples from patients with ASCC used for ctHPV16 testing and disease monitoring.

**Figure 2 cancers-15-00867-f002:**
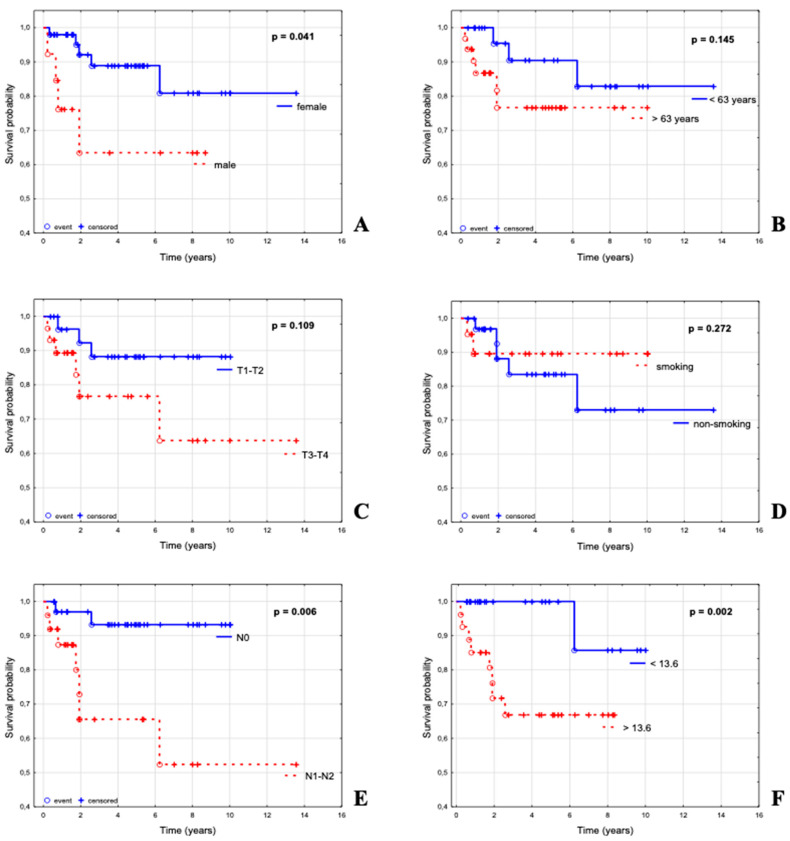
Overall survival (OS) of 62 patients presented using Kaplan–Meier curves stratified by sex (**A**), age (**B**), T stage (**C**) smoking (**D**), N stage (**E**), SUVmax (**F**).

**Figure 3 cancers-15-00867-f003:**
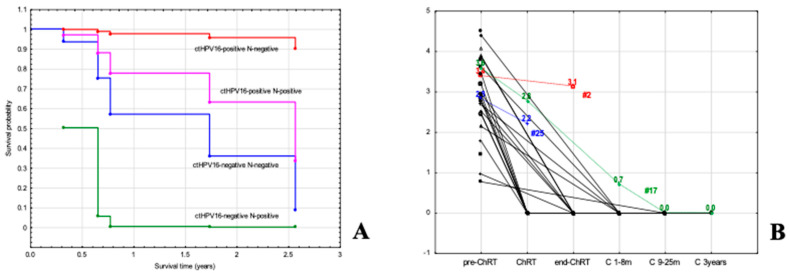
(**A**) Cox proportional hazard model for N and ctHPV16 status in patients with large SUVmax (>13.6). (**B**) Monitoring of ctHPV16–positive patients.

**Table 1 cancers-15-00867-t001:** Prognostic impact of selected parameters on overall survival of 62 patients with ASCC treated with CRT.

Parameter	Univariate Analysis		Multivariate Analysis	
	HR (95% CI)	*p* Value	HR (95% CI)	*p* Value
Female vs. male (ref.)	0.29 (0.08–1.07)	0.063		
Age at diagnosis (cont.)	1.05 (0.97–1.14)	0.189		
T1/T2 vs. T3/T4 (ref.)	0.34 (0.08–1.37)	0.128	0.20 (0.03–1.52)	0.119
N–negative vs. N–positive (ref.)	0.14 (0.03–0.69)	0.016	0.14 (0.02–0.76)	0.023
non–smoker vs. smoker (ref.)	1.39 (0.27–7.20)	0.692		
SUVmax	1.98 (0.98–4.00)	0.057	4.40 (1.29–15.06)	0.018
I ≤ 9.61;				
II 9.61–13.6;				
III 13.6–17.4;				
IV ≥ 17.4				
(cont.)				

For the multivariate analysis, all parameters from the univariate analysis were taken, (ref.)—indicates a reference parameter, (cont.)—continuous variable, HR—hazard ratio, CI—confidence level.

**Table 2 cancers-15-00867-t002:** Comparative analysis according to blood–based ctHPV16 status (column B–D). VL of ctHPV16 positive patients with ASCC (N = 27) (columns F–H) presented in crude data (column F) or log_10_ (column G).

	A	B	C	D	F	G	H
Parameter	All	ctHPV16-Negative (N = 8)	ctHPV16-Positive (N = 27)	*p* Value	VL in Copies/mL	Log_10_ VL	*p*-Value
sex							
Female	28 (80%)	5 (14%)	23 (66%)	0.158	1502 (6–31,500)	3.2 (0.8–4.5)	0.452
Male	7 (20%)	3 (9%)	4 (11%)		665 (277–895)	2.8 (2.4–3.0)	
age							
age < 63 (median)	14 (40%)	4 (11%)	10 (29%)	0.511	2029 (60–11,800)	3.3 (1.8–4.1)	0.152
age ≥ 63 (median)	21 (60%)	4 (11%)	17 (49%)		485 (6–31,500)	2.7 (0.8–4.5)	
T classification							
1	2 (6%)	1 (3%)	1 (3%)	0.664	287 (N/A)	2.5 (N/A)	(T1/T2 vs. T3/T4) 0.836
2	13 (39%)	2 (6%)	11 (33%)		1502 (6–7750)	3.2 (0.8–3.9)	
3	17 (52%)	4 (12%)	13 (39%)		672 (9–31,500)	2.8 (1.0–4.5)	
4	1 (3%)	0 (0%)	1 (3%)		662 (N/A)	2.8 (N/A)	
N classification							
0	20 (57%)	5 (14%)	15 (43%)	0.905	844 (6-7380)	2.9 (0.8–3.9)	N-negative vs. N-positive 0.317
1a	11 (31%)	2 (6%)	9 (26%)		895 (60–31,500)	3.0 (1.8–4.5)	
1c	4 (11%)	1 (3%)	3 (9%)		672 (277–25,167)	2.8 (2.4–4.4)	
cigarette consumption							
non–smoker	18 (56%)	3 (9%)	15 (47%)	0.732	1502 (9–25,167)	3.2 (1.0–4.4)	0.436
smoker	14 (44%)	3 (9%)	11 (34%)		585 (6–31,500)	2.8 (0.8–4.5)	
SUVmax							
≤9.61	9 (28%)	2 (6%)	7 (22%)	0.386	1598 (6–11,800)	3.2 (0.8–4.1)	0.606
9.62–13.60	7 (22%)	1 (3%)	6 (19%)		408 (140–31,500)	2.6 (2.1–4.5)	
13.61–17.40	7 (22%)	0 (0%)	7 (22%)		2460 (277–25,167)	3.4 (2.4–4.4)	
>17.41	9 (28%)	3 (9%)	6 (19%)		1087 (9–7750)	3.0 (1.0–3.9)	

One case with TX (primary tumor cannot be assessed) had 2840 (log 3.4) copies/mL.

**Table 3 cancers-15-00867-t003:** Analysis of disease-free survival (DFS) and overall survival (OS) in the subgroup of ASCC patients treated with CRT and known pre-treatment status of ctHPV16 (N = 35).

	Disease-Free Survival (DFS)				Overall Survival (OS)			
	Univariate Analysis		* Multivariate Analysis		Univariate Analysis		* Multivariate Analysis	
	HR (95% CI)	*p* Value	HR (95% CI)	*p* Value	HR (95% CI)	*p* Value	HR (95% CI)	*p* Value
female vs. male (ref.)	0.999 (0.11–8.96)	0.999			0.25 (0.04–1.60)	0.142		
age at diagnosis (cont.)	1.00 (0.93–1.08)	0.861			1.00 (0.92–1.08)	0.987		
T1/T2 vs. T3/T4 (ref.)	0.31 (0.04–2.81)	0.301			0.53 (0.08–3.33)	0.496		
N–negative vs. N–positive (ref.)	1.06 (0.18–6.37)	0.946			0.31 (0.05–1.99)	0.216		
non–smoker vs. smoker (ref.)	0.25 (0.03–2.38)	0.226			0.61 (0.08–4.37)	0.619		
SUVmax (I ≤ 9.61; II 9.61–13.6; III 13.6–17.4; IV ≥ 17.4) (cont.)	1.80 (0.75–4.31)	0.185			5.08 (0.80–32.29)	0.084	6.29 (1.02–38.62)	0.047
ctHPV16–negative vs ctHPV16–positive (ref.)	2.25 (0.37–13.50)	0.374	5.30 (0.74–37.71)	0.096	2.20 (0.37–13.15)	0.389	38.27 (0.94–1555.0)	0.054

* for the multivariate analysis, all parameters from the univariate analysis were taken, (ref.)—indicates a reference parameter, (cont.)—continuous variable, HR—hazard ratio, CI—confidence level.

## Data Availability

The datasets used and/or analysed during the current study are available from the corresponding author on reasonable request.

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
