# Peer review of "Circulating HPV16 DNA in Blood Plasma as Prognosticator and Early Indicator of Cancer Recurrence in Radio-Chemotherapy for Anal Cancer"

_cancers, 2023, doi:10.3390/cancers15030867_

Round 1

Reviewer 1 Report

With pleasure, I read the article titled “Circulating HPV16 DNA in Blood Plasma as Prognosticator and Early Indicator of Cancer Recurrence in Radio-Chemotherapy for Anal Cancer”. The article is clinically relevant and within the scope of the journal. Overall, the article reads well, the English language is proper, citations are adequate, and flow of ideas is smooth. The presented summaries in the form of figures and tables are major strengths. Overall, a well-done job. I just have a few suggestions:

(1) Abstract. Please double-check again — “Results: In the multivariate Cox analysis, lower SUVmax (p=0.047) and ctHPV16-positive (p=0.054) proved to be independent prognostic factors for poor OS”.

(2) Introduction. Please spell out all abbreviations upon first encounter. The authors need to clearly highlight the gap in literature. Also, the authors need to clearly highlight the significance of this work and specify what exactly it contributes to the existing literature. Lastly, the authors may want to end up the section with a hypothesis.

(3) Methods. The statistical analysis section is missing some information. What is the rationale for using the Mann-Whitney U test? Is it because the data was not normally distributed, and whether the Kolmogorov-Smirnov or Shapiro-Wilk tests, for example, were used to test this aspect? Were the p-values one- or two-tailed? How was statistical significance determined— is it p value <0.05? How was overall survival (OS) defined in this study?

(4) Results. If available, I would encourage enriching the baseline characteristics of patients, such as body mass index, history of multiple sexual partners, history of cancers, history of anal sex, history of immune suppression, etc. it is very important to understand the risk factors in your cohort. For Table 1, for sex, please specify the order of gender comparison. Also, why the gender, age at diagnosis, and use of tobacco were not entered into the multivariate cox regression analysis? For all the Tables, please indicate whether the parameter was evaluated as numerical or categorical and please highlight the control parameter or order of comparison (for example male vs female)? For Table 3, although SUVmax was not statistically significant during univariate analysis, however, it was statistically significant during the multivariate analysis; why? For disease relapse, it would be better if the authors examined the progression-free survival (PFS) or disease-free survival (DFS) rates along with the overall survival (OS). These data can be used to complement the utility of ctHPV16 DNA.

(5) Discussion. This statement is bold and needs clarification: “In our study we underscore that, ctHPV16 appears to have a greater impact on prognosis than N status”. HPV 16 and HPV 18 are somehow largely related and equally common. Where HPV18 related ASCC cases excluded in your study? Are there published literature that have investigated the role of ctDNA HPV18 in ASCC? Please aknowldge the limitations of your study (for example, retrospective study, small sample size, single-center experience, etc).

(6) Overall. The manuscript reads well. The references are appropriate. The English language is proper, but it will benefit from minor polishing.

Author Response

Authors: I would like to thank the Reviewer for this kind opinion and thorough analysis of the presented manuscript, which in my opinion significantly improved the manuscript. Below are the answers.

With pleasure, I read the article titled “Circulating HPV16 DNA in Blood Plasma as Prognosticator and Early Indicator of Cancer Recurrence in Radio-Chemotherapy for Anal Cancer”. The article is clinically relevant and within the scope of the journal. Overall, the article reads well, the English language is proper, citations are adequate, and flow of ideas is smooth. The presented summaries in the form of figures and tables are major strengths. Overall, a well-done job. I just have a few suggestions:

(1) Abstract. Please double-check again — “Results: In the multivariate Cox analysis, lower SUVmax (p=0.047) and ctHPV16-positive (p=0.054) proved to be independent prognostic factors for poor OS”.

Authors: we corrected to “favorable overall survival (OS)”

(2) Introduction. Please spell out all abbreviations upon first encounter. The authors need to clearly highlight the gap in literature. Also, the authors need to clearly highlight the significance of this work and specify what exactly it contributes to the existing literature. Lastly, the authors may want to end up the section with a hypothesis.

Authors: We checked all the abbreviations and expanded them in the first expression used.

Authors: Introduction: To highlight the gap in literature, significance of this work and hypothesis we have inserted the following excerpt:

Most of the studies on prognostic significance of HPV infection in anal cancer referred to results based on the analysis of tumor biopsy samples [12]. Studies on ctHPV16 (assessed for the blood) are still few and fragmentary [11,13-15], The researchers report that there was no difference in survival time between ctHPV-positive and ctHPV-negative patients (in pre-treatment samples). It was shown that only dividing by viral load allowed patients to be stratified in terms of survival prognosis [14,15]. Of note, the ability of early assessment of disease relapse based the analysis ctHPV16 is not robustly established due to rarity of disease and relatively low rates of relapses. There is a large gap in publications on the importance of ctHPV in anal cancer, while our prospective cohort study addresses several threads, among them survival, monitoring, multivariable regression.

The hypothesis is that ctHPV16 derived from blood plasma may be useful for monitoring treatment effects, disease relapse and prognostic marker in curative radio-chemotherapy for anal cancer. Also, we hypothesize that it is possible to define, based on ctHPV16 analysis, clinical subgroups of low-risk patients who may, potentially, benefit from de-intensification of therapy and high-risk group who may candidate for more aggressive treatment.

(3) Methods. The statistical analysis section is missing some information. What is the rationale for using the Mann-Whitney U test? Is it because the data was not normally distributed, and whether the Kolmogorov-Smirnov or Shapiro-Wilk tests, for example, were used to test this aspect? Were the p-values one- or two-tailed? How was statistical significance determined— is it p value <0.05? How was overall survival (OS) defined in this study?

Authors: We used the Mann-Whitney test to assess the differences between the groups because it is dedicated to the comparison of small-sized samples with an abnormal distribution. In addition, this test has been used in most of the work in our field, allowing us (or the target of future meta-analysis) to easily compare the results obtained. Of course, we have adapted to the following test requirements: measurement on a continuous level, the independent variable consisted of two categorical, independent observations, abnormal distribution, small number in the compared groups.

We added to the text: (two-tailed test)

We added to the text: A p-value of less than 0.05 was considered to be statistically significant.

We added to the text: Overall survival (OS) rate was defined as the percentage of individuals in the group who survived after treatment. Disease-free survival (DFS) was defined as the length of time after treatment ends that the patient survives without any signs or symptoms of that cancer.

(4) Results. If available, I would encourage enriching the baseline characteristics of patients, such as body mass index, history of multiple sexual partners, history of cancers, history of anal sex, history of immune suppression, etc. it is very important to understand the risk factors in your cohort. For Table 1, for sex, please specify the order of gender comparison.

Authors: As suggested by reviewer , we reviewed the patients' medical records again. Since we felt that they might be important to the reader, we have included the following:

We added subchapter 3.1 Patients' characteristics. Next, we added to the subchapter 3.1: Five patients had been diagnosed with another malignancy between 6 and 21 years prior to treatment, 1 patient had breast cancer and colorectal cancer, 2 had cervical cancer, 1 had oropharyngeal cancer, 1 had breast cancer , 2 patients had genital warts. None of the patients received immunosuppressive therapy before or during treatment. The data about patients' sexual life were not available.

Also, why the gender, age at diagnosis, and use of tobacco were not entered into the multivariate cox regression analysis?

We added to the text (below the tables): for the multivariate analysis, all parameters from the univariate analysis were taken

For all the Tables, please indicate whether the parameter was evaluated as numerical or categorical and please highlight the control parameter or order of comparison (for example male vs female)?

Authors: According to the Reviewer's suggestion, changes were made to the parameter columns. For discrete parameters, the abbreviation "ref." is used, which means a reference parameter. Parameters used as continuous are marked with the abbreviation "cont".

Below the tables we provided the following explanations: “*for the multivariate analysis, all parameters from the univariate analysis were taken, (ref.) - indicates a reference parameter, (cont.) - continuous variable, HR – hazard ratio, CI - confidence level”.

For Table 3, although SUVmax was not statistically significant during univariate analysis, however, it was statistically significant during the multivariate analysis; why?

Authors:

In the manuscript, in Cox proportional hazards, we used backward elimination methods. Backward elimination is to start with a full model, then repeatedly removing one of the covariates from the model, selecting the best of these models, and then repeating this until all covariates are exhausted this Eliminations are done automatically. Leaving (automatic) variables, for example 2 variables, means that both together affect survival (e.g. HPV-negative with advanced nodes prognosis is poor). Variables affect the calculation of P, so a different value is obtained than in the univariate test.

We added to the text: The backward elimination method was used for the selection of variables in the multivariate analysis of the Cox proportional hazards model.

For disease relapse, it would be better if the authors examined the progression-free survival (PFS) or disease-free survival (DFS) rates along with the overall survival (OS). These data can be used to complement the utility of ctHPV16 DNA.

Authors: As suggested, an additional DFS analysis was performed, interestingly, only ctHPV was indicated as important (although not significantly) in the prognosis of cure. The analysis results are shown in the added columns of Table 3.

We added to the text, in connection with the conducted DFS analysis: Positive ctHPV16 test result was a good prognostic factor for DFS, although without significance (p = 0.096, Table 3).

Authors: An additional change was made during the correction of sentence “Sex, age, N and ctHPV16 status were taken for multivariate analysis”. My (AM) mistake, T was not included in the analysis. "T" (T1/2 vs T3/4) was not included, because multivariate was performed for both the subgroup with a larger SUVmax (results below and Figure 3A) and the subgroup T3/4 (we wrote: “Similar results we obtained for the analysis for advanced tumors (T3-T4), although ctHPV16 did not reach significance (p=0.057 for ctHPV16 and p=0.256 for N)”.

(5) Discussion. This statement is bold and needs clarification: “In our study we underscore that, ctHPV16 appears to have a greater impact on prognosis than N status”.

We added to the text: Our work sheds new light on the importance of nodal advancement in ASCC. It is possible that the high nodal stage in ctHPV-positive tumors may be the result of HPV infection. This phenomenon is highlighted by research on the OPC. Enlarged lymph nodes in the neck are very common in patients with HPV-positive oropharyngeal cancers [30]. Another interesting clinical observation is the high nodal grade in patients with small tumors (T1-T2) [31,32]. Through our study, we point out that HPV should be included in the stratification of patients with ASCC (as it happened in OPC in 2018), because HPV, by modifying the N feature, strongly influences the survival prognosis.

HPV 16 and HPV 18 are somehow largely related and equally common. Where HPV18 related ASCC cases excluded in your study? Are there published literature that have investigated the role of ctDNA HPV18 in ASCC?

Authors: In our study, we focused solely on HPV16 subtype monitoring. There is general agreement that HPV16 is the most prevalent type of HPV in anal cancer (Lin, meta-analysis 2018). Studies on the prevalence of individual HPV subtypes in anal cancer have shown that the dominant HPV subtype is HPV16 with a frequency of up to 95%, while the prevalence of the second most common HPV18 is around 5% (Baricevic, 2015 10.1016/j.ejca.2015.01.058  – 89% vs 5%, Cabel , 2018 10.1158/1078-0432.ccr-18-0922 – 95% vs 5%, Lefèvre, 2021 10.3390/cancers13102451  – 90% vs 4%). In comparison, in cervical cancer, the detection rate of HPV16 is estimated at 67%, while for HPV18 the rate is 16% (data for the European population, Crosbie, 2013 10.1016/S0140-6736(13)60022-7 ). More importantly, a comparison of the incidence of high-risk HPV subtypes in normal tissues - precancerous stages - anal cancer shows that only the incidence of HPV16 increases with the severity of the lesion, providing evidence for HPV16 carcinogenicity (Lin, 2018 10.1016/S1473-3099(17)30653-9 ). The strong carcinogenic effect of HPV16 also confirms the relationship between the presence of HPV16-positive anal lesions and HPV16-positive various grades of cervical lesions. The prevalence of HPV16-positive anal HSIL was almost 25% higher in women with cervical HPV16 infection than in HPV-negative women, compared with a prevalence rate of 4.6 in other high-risk non-HPV16 HPV types (Lin, 2019 10.1016/S1473-3099(19)30164-1). As for examining the role of HPV18 ctDNA in ASCC and using the presence of HPV18 to monitor the disease, due to the small number of HPV18-positive cases, they are usually included in the pool of HPV16-positive cases and analyzed together (Cabel , 2018 10.1158/1078-0432.ccr-18-0922, Lefèvre, 2021 10.3390/cancers13102451 ).

Please aknowldge the limitations of your study (for example, retrospective study, small sample size, single-center experience, etc).

We added to the text: Our study has several limitations. The study was retrospective. The sample wass relatively small and a small number of events occurred. In addition, the study was conducted in a single institution. Since the literature on the subject is quite limited, this paper aims to provide new data on this topic. We attempted to define a clinical subgroup of low-risk patients who may potentially benefit from de-intensification of therapy, and a high-risk group who may be eligible for more aggressive treatment. Through our multiple regression studies, we also showed that ctHPV16 is not the only significant predictor, as it should be noted that node staging is also of particular importance for larger tumors.

(6) Overall. The manuscript reads well. The references are appropriate. The English language is proper, but it will benefit from minor polishing.

Authors: Thank you very much, we have addressed as many of the dilemmas presented in the review as possible, conducted additional analysis and made changes.

Reviewer 2 Report

This single institution, retrospective study looking at 62 patients treated for anal cancer and predictors of treatment response and survival.

1.     Methods in abstract should include SUV as this is reported in the results and conclusions.

2.     I don’t understand the timing of blood draws, especially in the context of Figure 1. The labeling of Figure 1 is confusing as what numbers indicate ctDNA. This does not indicate if prior to treatment, after treatment and if after treatment when.

3.     They discuss the use of ctDNA, but only 35 patients are in this analysis. State associated with worse prognosis, but not statistically significant.

4.     I don’t understand the statement that this was a relatively homogenous group in the discussion.

5.     I don’t see what this study provides in terms of new findings compared to the established literature.

Author Response

Authors: Thank you very much for your valuable comments, we have introduced all the suggested corrections, hoping that they will significantly improve the manuscript.

This single institution, retrospective study looking at 62 patients treated for anal cancer and predictors of treatment response and survival.

  1. Methods in abstract should include SUV as this is reported in the results and conclusions.

Authors: corrected, in abstract we added:

“A threshold of 2.5 was used to determine the SUVmax”

  1. I don’t understand the timing of blood draws, especially in the context of Figure 1. The labeling of Figure 1 is confusing as what numbers indicate ctDNA. This does not indicate if prior to treatment, after treatment and if after treatment when.

We added to the text: Blood samples were collected from patients during routine tests before treatment to estimate HPV status (n=35), who were then monitored following chemoradiotherapy for up to 3 years. We also included 27 patients, who have not been tested for HPV before treatment in the post-treatment monitoring. The available samples and scheme of patient classification are summarized in Figure 1.

Authors: The layout in Figure 1 has been rearranged slightly (lowered box) to show that 27 patients were not sampled before treatment, therefore we do not have ctHPV16 results. The exact monitoring time points are presented in Sections 3.5. However, monitoring is presented separately in 2 sections: "Monitoring patients with ctHPV16 tested before treatment" and "Testing ctHPV16 in patients without a pre-treatment sample".

In subchapter " Monitoring of Patients with Tested ctHPV16 Prior to Treatment" figure 3B shows (results with time points) only these 27 HPV+ among 35. The negative is not shown in the diagram as it would just be a flat line.

In subchapter “Determination of ctHPV16 in Patients without Sample Collection before Treatment.”: For patients with no known pre-treatment ctHPV16 status, ctHPV16 was performed at various post-treatment points. The results in this group show that a negative ctHPV result correlated with the absence of symptoms of the disease, and the detected ctHPV16 in one patient was confirmed by relapse.

The following snippet has been inserted in the methods:

Of the 62 patients, 35 were tested for ctHPV16 in their blood prior to treatment and the results for this group are presented in Sections 3.3-3.5, including Section 3.5 on monitoring.

  1. They discuss the use of ctDNA, but only 35 patients are in this analysis. State associated with worse prognosis, but not statistically significant.

Authors: That's right, unfortunately squamous cell carcinoma of the anus is a very rare cancer. Annually, there are about 300 cases in Poland, and taking into account the number of oncology centers, 1 case per month is a great success. It is certainly noteworthy that HPV is a stronger predictor of a better cure than N. And N is a rather "harmless" result of "infectivity". At least this is the picture we see in oropharyngeal cancers.

We added to the text: Our work sheds new light on the importance of nodal advancement in ASCC. It is possible that the high nodal stage in ctHPV-positive tumors may be the result of HPV infection. This phenomenon is highlighted by research on the OPC. Enlarged lymph nodes in the neck are very common in patients with HPV-positive oropharyngeal cancers [30]. Another interesting clinical observation is the high nodal grade in patients with small tumors (T1-T2) [31,32]. Through our study, we point out that HPV should be included in the stratification of patients with ASCC (as it happened in OPC in 2018), because HPV, by modifying the N feature, strongly influences the survival prognosis.

  1. I don’t understand the statement that this was a relatively homogenous group in the discussion.

Authors: We meant that the group of patients has several differentiating classification of T or N, but is homogeneous in terms of treatment. The word "relatively" has been removed in the manuscript.

  1. I don’t see what this study provides in terms of new findings compared to the established literature.

Authors: To emphasize the importance of this work, we have inserted the following 2 fragments:

In Introduction:

Most of the studies on prognostic significance of HPV infection in anal cancer referred to results based on the analysis of tumor biopsy samples [12]. Studies on ctHPV16 (assessed for the blood) are still few and fragmentary [11,13-15], The researchers report that there was no difference in survival time between ctHPV-positive and ctHPV-negative patients (in pre-treatment samples). It was shown that only dividing by viral load allowed patients to be stratified in terms of survival prognosis [14,15]. Of note, the ability of early assessment of disease relapse based the analysis ctHPV16 is not robustly established due to rarity of disease and relatively low rates of relapses. There is a large gap in publications on the importance of ctHPV in anal cancer, while our prospective cohort study addresses several threads, among them survival, monitoring, multivariable regression.

In Discussion:

Our work sheds new light on the importance of nodal advancement in ASCC. It is possible that the high nodal stage in ctHPV-positive tumors may be the result of HPV infection. This phenomenon is highlighted by research on the OPC. Enlarged lymph nodes in the neck are very common in patients with HPV-positive oropharyngeal cancers [30]. Another interesting clinical observation is the high nodal grade in patients with small tumors (T1-T2) [31,32]. Through our study, we point out that HPV should be included in the stratification of patients with ASCC (as it happened in OPC in 2018), because HPV, by modifying the N feature, strongly influences the survival prognosis.

Round 2

Reviewer 1 Report

The authors did a very wonderful by addressing all the raised comments in details. The manuscript now reads very well, more solid, scientifically sound, and intellectually informative. The paper is acceptable for publication in its current format. A well-done revision!

Reviewer 2 Report

Authors have incorporated all suggested edits